# Electronic Properties and Structure of Silicene on Cu and Ni Substrates

**DOI:** 10.3390/ma15113863

**Published:** 2022-05-28

**Authors:** Alexander Galashev, Alexey Vorob’ev

**Affiliations:** Institute of High-Temperature Electrochemistry, Ural Branch, Russian Academy of Sciences, Sofia Kovalevskaya Str. 22, 620990 Yekaterinburg, Russia; vorobev@ihte.uran.ru

**Keywords:** adhesion, silicene, structure, substrate, lithium

## Abstract

Silicene, together with copper or nickel, is the main component of electrodes for solar cells, lithium-ion batteries (LIB) and new-generation supercapacitors. The aim of this work was to study the electronic properties and geometric structure of “silicene–Ni” and “silicene–Cu” systems intended for use as LIB electrodes. The densities of electronic states, band structures, adhesion energies and interatomic distances in the silicene–(Cu, Ni) systems were determined by ab initio calculations. Silicene on a copper substrate exhibited temperature stability in the temperature range from 200 to 800 K, while on a nickel substrate, the structure of silicene was rearranged. Adsorption energies and bond lengths in the “silicene–Cu” system were calculated in the range of Li/Si ratios from 0.125 to 0.5. The formation of the Li_2_ isomer during the adsorption of lithium in a ratio to silicon of 0.375 and 0.5 was observed. Silicene was found to remain stable when placed on a copper substrate coated with a single layer of nickel. The charge redistribution caused by the addition of a nickel intermediate layer between silicene and a copper substrate was studied.

## 1. Introduction

At present, the most efficient solar cells are based on pure (solar) silicon [1]. To create a metallic contact with a solar cell, a film composed of a highly conductive material is created on its outer surface [2]. For some time, silver served as an excellent conductive material, which increased significantly the cost of solar batteries [3]. Also Copper has high electrical conductivity and is much cheaper than silver. Naturally, copper was chosen as a silver substitute for coating solar cells. However, there have been significant obstacles to the use of copper for this purpose. The fact is that copper has high diffusion capacity and good solubility in Si. The direct deposition of copper on the surface of a solar cell, as a rule, leads to silicon contamination and subsequent degradation of the solar cell. To prevent the direct contact of Cu with the underlying Si, a thin layer of Ni is applied on the contact pads between Si and Cu. This layer is used as a Cu diffusion barrier [4]. Nickel silicide, formed on the contact surface [5], reduces the contact resistance between the metal and Si and also improves the adhesion of the metal to Si [6]. Thus, three chemical elements (Si, Ni and Cu) in direct contact with each other are involved in the creation of a relatively inexpensive and highly efficient solar battery. Thin plates of high-purity silicon with a bulk crystalline structure are used in solar cells. A two-dimensional silicon material, silicene, has a significantly higher electrical conductivity [7] than 3D crystalline silicon, and its mechanical properties are not inferior to those of solid bulk silicon [8]. Silicene, which has many unique properties, has already been proposed for use in various applications, resulting in a dramatic improvement of the performance of devices such as field-effect transistors [9] and lithium-ion batteries [10].

Nickel is one of the main components of electrodes for supercapacitors [11]. A high efficiency of supercapacitors is achieved due to its significant electrical conductivity and rapid ion diffusion. The metal substrate affects the electronic properties of the two-dimensional material. Thus, ab initio calculations showed that in graphene on a Ni (111) substrate, a band gap can open; the limiting value is ~400 meV [12]. Note that free-standing graphene has a zero-energy gap, while the silicene is a narrow-gap semiconductor with a 27 meV energy gap [13]. Copper has been used to obtain thin films of tin, including stanene [14]. In the design of a supercapacitor, it is proposed to use a double-sided electrode (anode/cathode) with a common current collector [15]. This electrode is made of a thin copper foil that carries deposited graphene in combination with a nickel foil. Replacing graphene with silicene in the anode of a lithium-ion battery will significantly increase the electrode capacity.

Currently, silicene is produced by epitaxial growth mainly on metal substrates. When choosing a substrate, it is necessary to take into account the presence of hexagonal symmetry in it, which allows the formation of silicene honeycombs. For this purpose, metals with a fcc surface (111) and metals with a hcp surface (0001) are suitable.

It is of interest to find out how silicene behaves in contact with thin films composed of nickel or copper when the thickness of these films changes. This is important for establishing if it is possible to use silicene as an active material for high-efficiency solar cells and anodes for new-generation lithium-ion batteries.

The purpose of this work was to study the stability, as well as the electronic and energy properties of silicene on nickel and copper plates, depending on the thickness of the metal substrate, based on quantum mechanical calculations.

## 2. Model

First-principle calculations were carried out on the basis of the density functional theory using the Siesta software package [16]. The exchange-correlation functional was defined within the generalized gradient approximation in Perdew–Burke–Ernzerhof form [17]. The Brillouin zone was considered according to the Monkhorst–Pack scheme [18] using 10 × 10 × 1 k points. All calculations were performed with a plane wave cutoff of 500 Ry. We used the Born–Karman periodic boundary conditions in the calculations. Geometric optimization was carried out for all considered “substrate–silicene” systems. The dynamic relaxation of atoms continued until the change in the total energy of the system became less than 10^−5^ eV.

Different silicene supercells were selected depending on the type of substrate. The silicene sheet was modeled on the basis of 2 × 2 supercells (eight atoms) located in the *xy* plane. The spatial translation period in the *z*-direction was 30 Å. The silicene structure was represented by two sublattices (lower and upper), separated from each other by a distance of 0.44 Å, which was received at work [19]. The layer of the metal substrate was specified as a 3 × 3 supercell consisting of nine metal atoms. The substrate thickness varied from one to five layers without fixing coordinates in a metal substrate. The SIESTA program uses the same translation period when combining cells into one supercell. In the case of Si and Cu, their cells coincide, with a 2% mismatch. This mismatch was eliminated by stretching the copper supercell and shrinking the silicene one by 1%. Figure 1 shows an example of combining supercells of 2 × 2 silicene and 3 × 3 copper. To obtain a combined Si–Ni supercell, a more significant correction of the Si–Si and Ni–Ni distances was required. The nickel supercell was enlarged by 4%, while the silicene supercell was decreased by 4%. As shown in [20,21], the 5%-reduced silicene superlattice retains its cellular structure, as well as its energy and electronic properties. Silicene–(copper substrate) systems with an intermediate layer of nickel between silicene and copper were considered. These systems were obtained by replacing the upper Cu layer of the substrate with one Ni layer (the substrate thickness varied from two to four copper layers).

A further increase in the thickness of the substrate layer was obtained by adding the lower layer fixed in the *z*-direction to the already existing non-fixed layers, followed by geometric optimization of this system. In addition, the thermal stability of the obtained systems, heated from 200 to 800 K with a temperature step of 100 K, was studied. Each calculation was carried out by the method of ab initio molecular dynamics (MD) in a Nose thermostat [22] during 1000 time steps with a step length of 1 fs.

When the optimal substrate thickness was determined, we simulated the adsorption of lithium in a ratio to silicene from 0.125 to 0.5. Lithium was deposited in the following way: first, above the center of the six-link ring, then over the silicon atoms of the silicene sheet. It was in this sequence that the adsorption of lithium on silicene was carried out in [23]. The resulting systems were subjected to geometric optimization, after which thermal stability was checked by the ab initio MD calculation using a Nose–Hoover thermostat at a temperature of 293 K. The computation time was 2000 time steps, the step length was 1 fs.

We calculated the adhesion energies *E*_adh_ between silicene and the substrates according to the formula presented below:(1)$$E_{\mathrm{adh}}=-\frac{E_{\mathrm{tot}}-E_{\mathrm{Si}}-E_{\mathrm{sub}}}{N_{\mathrm{el}}}$$
where *E*_tot_ is the total energy of the system, *E*_Si_ and *E*_sub_ are the total energies calculated for silicene and the metal (Ni, Cu) substrates, respectively, and *N*_el_ is the number of unit cells of silicene in the system.

The adsorption energy of lithium atoms on a silicene sheet applied on a copper substrate was calculated according to the expression:(2)$$E_{\mathrm{ads}}=-\frac{E_{\mathrm{tot}}-E_{\mathrm{Li}}-E_{\mathrm{SiCu}}}{N_{\mathrm{Li}}}$$
where *E*_SiCu_ and *E*_Li_ are energies calculated for the “silicene–copper substrate” subsystem and the lithium subsystem, respectively, and *N*_Li_ is the number of lithium atoms in the system.

The formula below was used to calculate the adhesion energies $E_{\mathrm{adh}}^{\mathrm{Ni}}$ between nickel and the copper substrates:(3)$$E_{\mathrm{adh}}^{\mathrm{Ni}}=-\frac{E_{\mathrm{tot}}-E_{\mathrm{SiNi}}-E_{\mathrm{Cu}}}{N_{\mathrm{el}}}$$
where, *E*_SiNi_ and *E*_sub_ are the total energies calculated for the silicene–nickel subsystem and the copper substrate, respectively.

## 3. Results and Discussion

The energy of adhesion is a determining indicator of the adhesion of a film to a substrate. This characteristic largely determines the durability of the thin film device used in electronic microcircuits. The magnitude of the adhesion forces, characterized by the adhesion energy, determines how quickly the film wears out, or, conversely, how easily the film can be removed from the substrate. In Appendix A, we determined by simulation the optimal thickness of a copper (nickel) substrate in a silicene–metal substrate system. The values obtained for the adhesion energies indicated that a four-layer substrate was sufficient for modeling the silicene–copper substrate and the silicene–nickel substrate systems. The transition from a three- to a four-layer substrate changed the adhesion energies between silicene and the metal substrate by 2.1 and 0.4% for the nickel and copper substrates, respectively. In further modeling, four-layer copper and nickel substrates were used, with the bottom layer of the substrate fixed in the *z* direction.

The geometric structures of silicene on copper and nickel substrates with a lower substrate layer fixed in the *z* direction are presented in Figure 2. The distance between the layers in the *z* direction (2.25 Å) is consistent with the cubic face-centered cell of these elements in the (111) direction. The values of adhesion energy of 2.21 and 2.87 eV between Si and Cu and Si and Ni, respectively, were obtained. These were rather high values of adhesion energy between metal and silicene. For example, in the case of contact between silicene and an Ag(111) substrate, the adhesion energy is 0.772 eV [24]. Changes were observed in the structure of silicene on the copper substrate. The silicene sheet became planar, with the exception of the edges, where some atoms were elevated above the others by 1.13 Å. The planar structure of silicene was also observed on an Au (111) substrate [13]. The violation of planarity at the edges of silicene was facilitated by the elevation of the extreme upper atoms of the copper substrate to a height of 0.53 Å. The upper layer of the nickel substrate had a different shape with respect to that of the corresponding layer of the copper substrate. In this case, the height of the buckles of silicene increased to ~0.85 Å, close to the values obtained for silicene on silver, 0.75 Å [25], and iridium, 0.83 Å, substrates [26].

The density of states (DOS) provides the ability to observe the possible underlying band singularity near the Fermi level. This singularity affects such bulk material properties as electrical conductivity. As a rule, conductivity is proportional to the lifetime of electrons at the Fermi surface. At a low concentration of charge carrier, these electrons’ lifetime turns out to be proportional to the DOS power law [27]. Thus, from the DOS shape in the vicinity of the Fermi level, we could conclude that the material was electronically conductive. Figure 3 shows the partial density of states of silicene on four-layer nickel and copper substrates. Silicene in these systems was metallized due to the interaction of p-electrons of silicon with d-electrons of nickel or copper. We also tracked the change in the electronic structure of silicene depending on the number of layers of the metal substrate. It turned out that, in all considered cases, regardless of the type of metal and the thickness of the substrate, silicene acquired conductive properties.

Figure 4 shows the initial geometric structure of silicene on a substrate (black circles) and its structure after ab initio molecular dynamics calculations. It can be seen that the copper systems under consideration were stable in the temperature range from 200 to 800 K. In none of the cases was either the separation of silicene from the substrates or the destruction of six-link rings in silicene observed. However, silicene destruction was observed on a nickel substrate over the entire considered temperature range. The destruction of a six-link silicon ring occurred because one silicon atom was displaced above the silicene plane, which resulted in the formation of a five-link silicon ring. The destruction of silicene occurred due to two main factors: 1. a strong interaction between silicene and the nickel substrate, which caused rearrangements in the silicene sheet; 2. the approximation used in the simulation, i.e., deformed supercells of silicene and nickel were used to simulate silicene on a nickel substrate.

It should be noted that the contact between silicene and the thin films composed of nickel and copper resulted in the shortening of the Si–Ni and Si–Cu distances. In the case of the copper substrate, the structure of silicene after the MD calculation was generally retained. However, on the nickel substrate, a structural rearrangement was observed, with the appearance of a five-membered ring instead of a six-membered one and the offset of some Si atoms to higher positions than the positions they occupied before performing the MD calculation. In addition, in contrast to the case of the interaction between bulk Si and Ni materials, the nickel atoms did not tend to approach the silicon surface, but rather, the Si atoms came close to the surface of the thin nickel film. This was due both to the small number of Si atoms forming silicene compared to the number of Ni atoms in the substrate, and to the fact that the Ni atoms are approximately 2.09 times heavier than the Si atoms. Cu atoms are 2.26 times heavier than Si atoms; however, the mean Si–Cu distance obtained by averaging the corresponding distances for substrates of different thicknesses turned out to be 7% larger than the mean Si–Ni distance. Note that the found average Si–Cu and Si–Ni distances were 5.4% greater and 1.6% smaller than the Si–Ni distance (2.438 Å) [28], obtained in the lowest energy structure of bulk nickel silicide, respectively. The reason for the shortening of the Si–Ni distance is the strong adhesion of silicene to nickel, which is ~6% stronger than that to copper [29].

The difference in the atomic radii of Cu and Ni is less than 3%. These metals exhibit the same electronegativity and valence state. The bulk phases of copper and nickel are completely miscible both in liquid and in solid states. The period of the FCC lattice of the Cu–Ni solid solution changes almost linearly with the atomic concentration of these elements.

Due to the significant structural changes observed in silicene in the ab initio MD modeling on the nickel substrate, we did not further consider this design as the LIB anode material. Therefore, we considered silicene on a copper substrate as an anode material and studied the change in the adsorption characteristics and structure of such material after lithium deposition.

An ab initio MD simulation of the adsorption of lithium on silicene on a copper substrate was carried out, in the ratio of lithium to silicon (*N*_Li_/*N*_Si_) from 0.125 to 0.5 at a temperature of 293 K. Table 1 shows the average bond lengths between silicon atoms and lithium ad-atoms (*L*_Si–Li_) and the adsorption energy of lithium on a silicene sheet (*E*_ads_). These characteristics change nonmonotonically as the number of adsorbed lithium atoms increases. The energy of a single lithium ad-atom adsorption is 0.753 eV, which is lower than that (2.082 eV) obtained for a single ad-atom adsorption on free-standing silicene in [23]. This difference is associated with the interaction of silicene with a copper substrate. The Si–Li bond length obtained for the adsorption of one lithium atom is higher than that (2.74 Å) obtained for a single adsorption of lithium on free-standing silicene [23]. However, an increase in the number of adsorbed lithium atoms leads to a decrease in the average bond lengths by 5–10%, until *N*_Li_/*N*_Si_ reaches 0.5.

Figure 5 and Figure 6 show the geometric structures of silicene systems on a copper substrate upon adsorption of one to four lithium atoms. It follows from these figures that the most probable position of a single lithium atom adsorption is above the center of a six-unit silicene ring, and in the considered range of lithium atoms adsorption (*N*_Li_/*N*_Si_ = 0.125, …, 0.5), no destruction of the silicene sheet was observed. In addition, the “silicene–copper substrate” system, after adsorbing from one to four lithium atoms, did not change its electronic properties, remaining a conductor. In the system corresponding to the adsorption indexes (*N*_Li_/*N*_Si_) of 0.375 and 0.5, the lithium isomer Li_2_ formed.

Appendix B shows the simulations of silicene on a one-, two-, and three-layer copper substrate with an intermediate layer of nickel in between. Such systems were stable at a temperature of 293 K. As the substrate thickness increased from two to three copper layers, the change in adhesion energy between the substrate and silicene was less than 0.1%. In the silicene–copper substrate systems with an intermediate nickel layer, the charge was redistributed. Nickel is more electronegative than silicon and copper. The electronegativity of nickel on the Pauling scale is 1.91, while the electronegativity of copper and silicon is 1.9. As a result, there was a shift of electron pairs from silicon and copper to nickel. This led to the flow of a negative charge from copper and silicene to nickel, and the resulting layers of opposite charges created additional electrostatic adhesion between silicene and nickel and between nickel and copper.

This study made it possible to answer some fundamental questions related to the implementation of first-principle molecular dynamics modeling and to provides a physical explanation for the process of formation of strong heterolayers. Indeed, on the one hand, we found that a four-layer thickness of a metal substrate is sufficient to reproduce the geometric and energy properties of the corresponding macrosystem. On the other hand, we showed that the formation of heterolayers can be affected by the redistribution of charges in the material.

## 4. Conclusions

In this work, we studied the effect of Ni(111) and Cu(111) substrates on the physicochemical properties of silicene. The study was carried out by the method of quantum mechanical modeling. The obtained spectra of the electronic states indicated the appearance of conductivity of silicene on metal substrates due to the interaction of p-electrons of silicon with d-electrons of copper or nickel. Each of the considered metal substrates (copper, nickel) caused geometric rearrangements in silicene inherent only to the particular metal. As soon as the thickness of the metal substrate reached three layers, the adhesion energy between silicene and the substrate became weakly dependent on the thickness of the substrate. Thermal instability of the silicene–nickel substrate system was revealed in the temperature range from 200 to 800 K. The adsorption of lithium on silicene deposited on a copper substrate was studied. A decrease in the adsorption energy was revealed during the adsorption of lithium on silicene on a copper substrate compared to the adsorption of lithium on a free-standing silicene sheet. The stability of the silicene sheet during the adsorption of lithium in the ratio to silicon in the system (*N*_Li_/*N*_Si_) from 0.125 to 0.5 was shown. The formation of the lithium isomer Li_2_ was found at ratios of lithium to silicon in the system of 0.375 and 0.5. It was established that in a “silicene–copper substrate” system, which includes one intermediate Ni layer, the Ni–Cu adhesion energy increases with an increase in the number of Cu layers in the substrate. The nickel layer in such system carries a negative charge, while silicene and the copper portion of the substrate, regardless of its thickness, are positively charged.

The results obtained in this work indicate the possibility of the effective use of silicene in structures having a copper current collector, for the anode of lithium-ion batteries, as well as for silicene–copper solar cells with an intermediate nickel layer used as a diffusion barrier.

## Figures and Tables

**Figure 1 materials-15-03863-f001:**
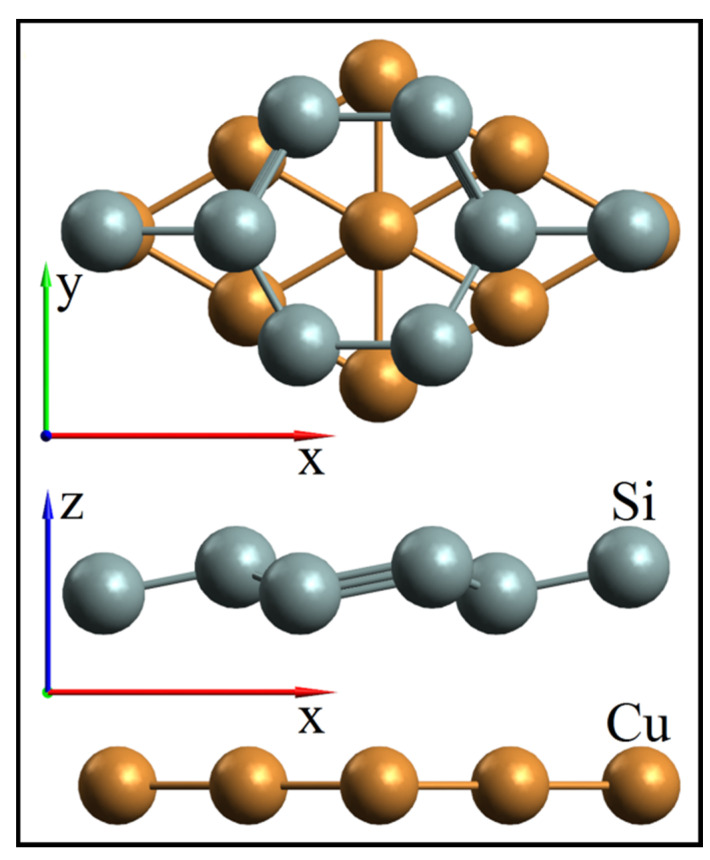
*xy* (**top**) and *xz* (**bottom**) projections of combined silicene and copper supercells.

**Figure 2 materials-15-03863-f002:**
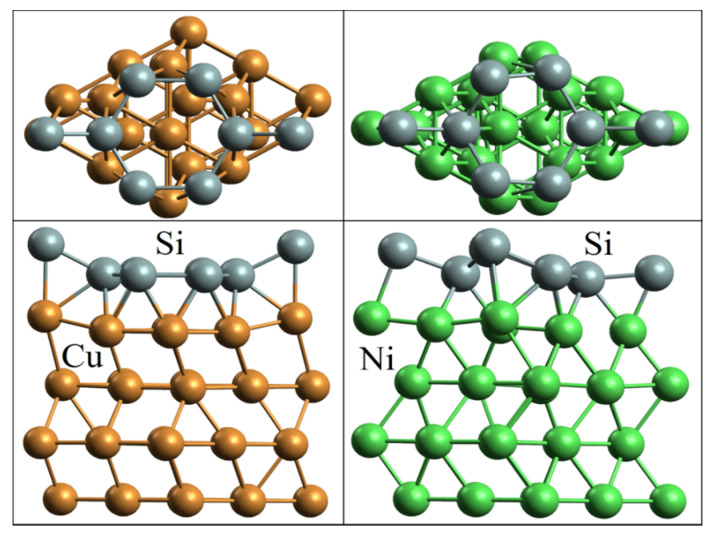
Geometrical structures of silicene on four-layer copper and nickel substrates after geometric optimization.

**Figure 3 materials-15-03863-f003:**
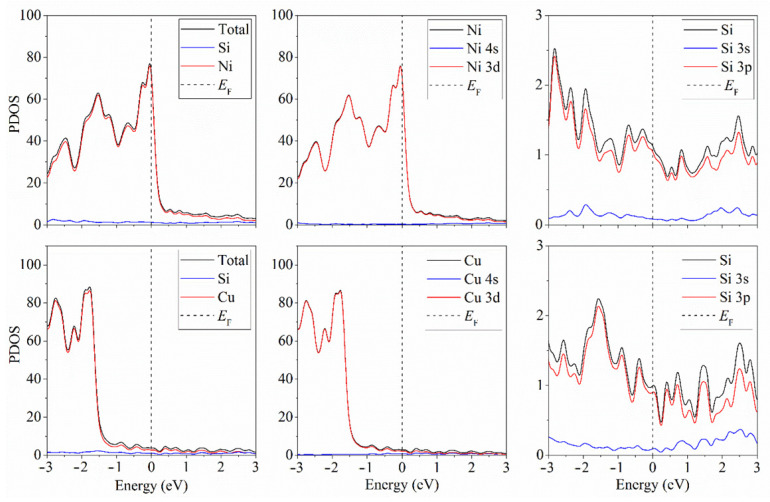
Partial density of states of the “silicene–four-layer nickel substrate” and “silicene–four-layer copper substrate” systems.

**Figure 4 materials-15-03863-f004:**
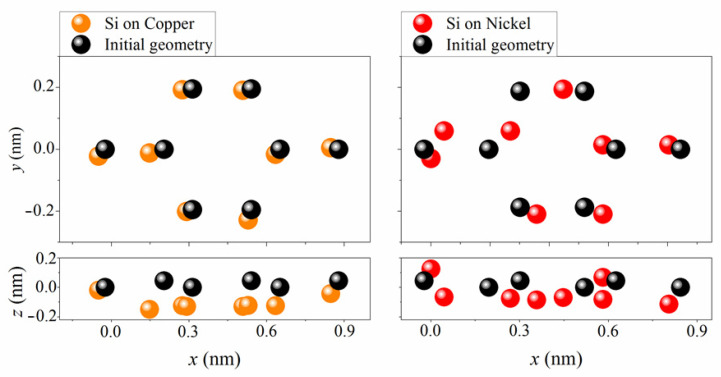
Geometric structures of silicene on four-layer copper and nickel substrates obtained by the time instant of 2000 ps in ab initio MD calculation at a temperature of 500 K (orange and red circles) and initial silicene structures (black circles).

**Figure 5 materials-15-03863-f005:**
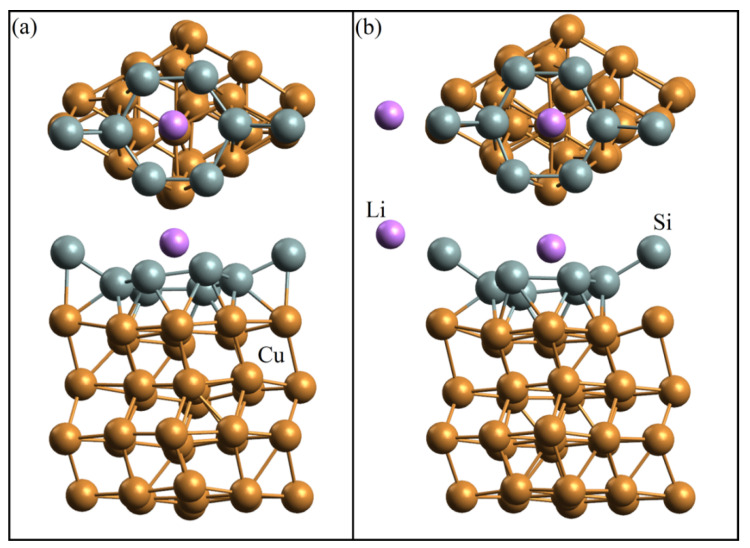
Geometric structures of silicene on the four-layer copper substrate with the addition of (**a**) one and (**b**) two Li atoms (after 2000 ps at a temperature of 293 K).

**Figure 6 materials-15-03863-f006:**
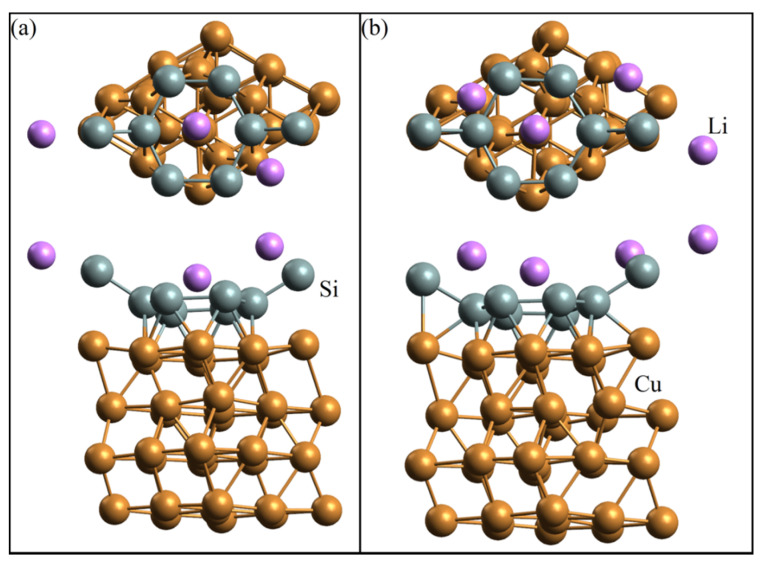
Geometric structures of silicene on the four-layer copper substrate with the addition of (**a**) three and (**b**) four Li atoms (after 2000 ps at a temperature of 293 K).

**Table 1 materials-15-03863-t001:** Average Si–Li length and adsorption energy of Li for different quantities of Li atoms in the system.

Property	*N*_Li_/*N*_Si_			
	0.125	0.25	0.375	0.5
*L* _Si–Li_	2.821	2.684	2.527	2.695
*E* _ads_	0.753	0.912	0.710	1.056

## Data Availability

The data presented in this study are contained within the article.

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
