# Peer review of "Electronic Properties and Structure of Silicene on Cu and Ni Substrates"

_materials, 2022, doi:10.3390/ma15113863_

Round 1

Reviewer 1 Report

Line 50. Change: ‘Copper can be used to’ with ‘Copper has been used to’

Line 52: Currency collector should be replaced with the current collector

Line 302: change number layers to the number of layers. Modify the whole sentence, the meaning is not clear

Line 306-307 needs to eb rewritten

Line 313: how can the Ni layer can carry a negative charge. Similarly, copper and silicine carried positive charge. What does this mean and what is the implication of it

Line 69: change functional to functional

Lines 77-81. Please provide some references.

Lines 85-86, practically how this can be done. Sam for lines 88-89

Lines 106-107, is it practically possible

Please explain the term metallization of silicone

The work should be correlated with the state-of-the-art studies presented in the literature. Try to compare the results so that the validity of the results can be established.

The manuscript ends abruptly, please summarize before the conclusion

Author Response

Line 50. Change: ‘Copper can be used to’ with ‘Copper has been used to’

The remark was taken into account.

Line 52: Currency collector should be replaced with the current collector

The remark was taken into account.

Line 302: change number layers to the number of layers. Modify the whole sentence, the meaning is not clear

This sentence was deleted during conclusion rewriting.

Line 306-307 needs to eb rewritten

The captions under figures 5 and 6 have been changed to the following:

Figure 5. Geometric structures of silicene on four-layer copper substrate with the addition of (a) 1 and (b) 2 Li atoms (after 2000 ps at a temperature of 293 K).

Figure 6. Geometric structures of silicene on four-layer copper substrate with the addition of (a) 3 and (b) 4 Li atoms (after 2000 ps at a temperature of 293 K).

Line 313: how can the Ni layer can carry a negative charge. Similarly, copper and silicene carried positive charge. What does this mean and what is the implication of it

The following additions have been made to the text:

In the silicene/copper substrate systems with an intermediate nickel layer, the charge is redistributed. Nickel is more electronegative than silicon and copper. The electronegativity of nickel on the Pauling scale is 1.91, while the electronegativity of copper and silicon is 1.9. As a result, there is a shift of electron pairs from silicon and copper to nickel. This electron displacement increases the interaction between silicene-nickel and nickel-copper due to electrostatic interaction. This leads to the flow of a negative charge from copper and silicene to nickel, and the resulting layers of opposite charges create additional electrostatic adhesion between silicene and nickel and between nickel and copper.

Line 69: change functional to functional

The remark was taken into account

Lines 77-81. Please provide some references.

Reference 19 has been added to the text:

The silicene structure was represented by two sublattices (lower and upper), separated from each other by a distance of 0.44 Å, which was observed in [19].

Lines 85-86, practically how this can be done. Sam for lines 88-89

Deformed cells were taken: cell deformation was achieved by increasing/decreasing translation vectors. These deformations are necessary for modeling these systems.

Lines 106-107, is it practically possible

The following additions have been made to the text:

It was in this sequence that the adsorption of lithium on silicene was carried out in [23].

Please explain the term metallization of silicone

Metallization of silicene in work was replaced by the appearance of conductivity:

The obtained spectra of electronic states indicate the appearance of conductivity of silicene on metal substrates due to the interaction of p-electrons of silicon with d-electrons of copper or nickel.

The work should be correlated with the state-of-the-art studies presented in the literature. Try to compare the results so that the validity of the results can be established.

The manuscript ends abruptly, please summarize before the conclusion

The following additions have been made to the text:

The study made it possible to answer some fundamental questions related to the implementation of first-principle molecular dynamics modeling and to give a physical explanation for the process of formation of strong heterolayers. Indeed, on the one hand, we found that a four-layer thickness of a metal substrate is sufficient to reproduce the geometric and energy properties of the corresponding macrosystem. On the other hand, we have shown that the formation of heterolayers can be affected by the redistribution of charges in the material.

Reviewer 2 Report

In the manuscript, adhesion energy and atom lengths have been calculated based on ab initio models for silicene covered with different number of atomic layers of Ni and Cu. The results show shortening of the atom bonds and variation of the atom locations caused by silicene – metal interactions. The conclusions cover the influence of the metals on the stability of the silicene structures and possibility to predict the adhesion parameters from the number of layers and metal nature. The authors did not provide many details of ab initio MD calculations but added the information on the similar effects of Li ad-atoms. The latter one is neither mentioned in the abstract nor discussed in conclusion. IT is obvious that this part of the work is related to the application of silicene-metal composites in lithium batteries but the authors should either extend this part of the work or remove the Li containing systems from consideration.

Author Response

In the manuscript, adhesion energy and atom lengths have been calculated based on ab initio models for silicene covered with different number of atomic layers of Ni and Cu. The results show shortening of the atom bonds and variation of the atom locations caused by silicene – metal interactions. The conclusions cover the influence of the metals on the stability of the silicene structures and possibility to predict the adhesion parameters from the number of layers and metal nature. The authors did not provide many details of ab initio MD calculations but added the information on the similar effects of Li ad-atoms. The latter one is neither mentioned in the abstract nor discussed in conclusion. IT is obvious that this part of the work is related to the application of silicene-metal composites in lithium batteries but the authors should either extend this part of the work or remove the Li containing systems from consideration.

Part of the work has been moved to Appendix A and Appendix B.

A sentence has been added to the abstract:

The formation of the Li2 isomer during the adsorption of lithium in a ratio to silicon of 0.375 and 0.5 was found.

The following has been added to the results:

An ab initio MD simulation of the adsorption of lithium on silicene on a copper substrate was carried out in the ratio of lithium to silicon (NLi / NSi) from 0.125 to 0.5 at a temperature of 293 K.

In the system corresponding to the adsorption indexes (NLi / NSi) of 0.375 and 0.5, the lithium isomer Li2 is formed.

Conclusion has been extended:

The stability of the silicene sheet during the adsorption of lithium in the ratio to silicon in the system (NLi / NSi) from 0.125 to 0.5 is shown. The formation of lithium isomer Li2 was found at ratios of lithium to silicon in the system of 0.375 and 0.5.

Reviewer 3 Report

The article presents first principles simulations of silicene on Ni(111) and Cu(111) surfaces. The adsorption energy of Li atoms on silicene/Cu(111) system is also reported.

I have the following remarks.

1) The adhesion energy and bond lengths are computed and discussed as a function of the number of layers of the substrate (lines 128-156 and tab.1). The authors do not state the reasons of this study. If the authors performed this calculation to establish the number of layers necessary to obtain results for the adhesion energy of silicene layers that does not depend on the thickness of substrate slab. i.e., to obtain data that mimic the silicene on a thick substrate as the one that can be found in an experiment, these simulations are simply a test, and these data should be presented in an appendix or in a test section after the model section.

In this case, only data that refers to the four-layer nickel and copper substrate should be reported in the section results (the four-layer substrate is the one chosen to compute the DOS and the PDOS).

This will improve the readability of the paper. If, on the contrary, the authors associate a physical meaning to the data for one, two, or three layers of Ni or Cu this meaning should be clearly discussed in the results section.

Physical results, i.e., properties that can be associated with effective physical systems that can be detected in an experiment should be clearly distinguished from tests performed to ensure reliable computational data reproducing physical quantities.

2) The same remarks can be extended to the study of “the geometric structures of silicene system on cupper substrate of various thicknesses with an intermediate layer of nickel” (Fig.5 tab.2 lines 231-256)

Further remarks:

3) Line 159: “The geometric structure of silicene on copper and nickel substrate with fixed lower substrate layer are presented in Fig.2”. Is it means that the choice of the lattice in-plane (i.e., in the plane parallel to the surface) the lattice parameter of the substrate is keep fixed? what is its value? how was it chosen?

4) In the introduction the authors stated: “copper has a high diffusion capacity and good solubility in Si” (line 29).” To prevent direct contact of Cu with the underlying Si, a thin layer of Ni is applied on the contact pads between Si and Cu.” (lines 32,33). This motivated the study in the paper of silicene system on cupper substrate with an intermediate layer of nickel. However, the authors found “significant structural changes observed in silicene in ab initio MD modeling on a nickel substrate” (line 263) for the silicene sheet and a nickel layer on copper substrate. In view of these finding, can the authors explain how they can conclude that “…the work indicates the possibility of effective use of silicene … for a silicene/copper solar cells with an intermediate nickel layer used as a diffusion barrier” (line 317). If silicene degrade on a Ni monolayer on Cu(111) it seems not possible to use silicene in such a system.

The paper cannot be published in the present form and significant revisions should be made before it will be further considered for publications.

Author Response

1) The adhesion energy and bond lengths are computed and discussed as a function of the number of layers of the substrate (lines 128-156 and tab.1). The authors do not state the reasons of this study. If the authors performed this calculation to establish the number of layers necessary to obtain results for the adhesion energy of silicene layers that does not depend on the thickness of substrate slab. i.e., to obtain data that mimic the silicene on a thick substrate as the one that can be found in an experiment, these simulations are simply a test, and these data should be presented in an appendix or in a test section after the model section.

In this case, only data that refers to the four-layer nickel and copper substrate should be reported in the section results (the four-layer substrate is the one chosen to compute the DOS and the PDOS).

This will improve the readability of the paper. If, on the contrary, the authors associate a physical meaning to the data for one, two, or three layers of Ni or Cu this meaning should be clearly discussed in the results section.

Physical results, i.e., properties that can be associated with effective physical systems that can be detected in an experiment should be clearly distinguished from tests performed to ensure reliable computational data reproducing physical quantities.

The modeling of the substrate thickness has been moved to Appendix A. A discussion of the results obtained and a reference to Appendix A have been added to the text:

The energy of adhesion is a determining indicator of the adhesion of the film to the substrate. This characteristic largely determines the durability of the thin film device used in electronic microcircuits. The magnitude of the adhesion forces, characterized by the adhesion energy, determines how quickly the film wears out, or, conversely, how easily the film can be removed from the substrate. In appendix A, we dedicated by simulation the optimal thickness of the copper/nickel substrate in the silicene-metal substrate system. The values obtained for the adhesion energies indicate that a 4-layer substrate is sufficient for modeling the silicene-copper substrate and silicene-nickel substrate systems. The transition from 3 to 4 layer substrate changes the adhesion energies between silicene and metal substrate by 2.1 and 0.4% for nickel and copper substrate, respectively. In further modeling, 4-layer copper and nickel substrates were used with the bottom layer of the substrate fixed in the z direction.

2) The same remarks can be extended to the study of “the geometric structures of silicene system on cupper substrate of various thicknesses with an intermediate layer of nickel” (Fig.5 tab.2 lines 231-256)

The modeling of the silicene on a one-, two-, and three-layer copper substrate with an intermediate layer of nickel between silicene and substrate has been moved to Appendix B. A discussion of the results obtained and a reference to Appendix B have been added to the text:

Appendix B shows simulations of silicene on a one-, two-, and three-layer copper substrate with an intermediate layer of nickel between silicene and substrate. Such systems are stable at a temperature of 293 K. As the substrate thickness increased from two to three copper layers, the change in the adhesion energy between the substrate and silicene was less than 0.1%. In the silicene/copper substrate systems with an intermediate nickel layer, the charge is redistributed. Nickel is more electronegative than silicon and copper. The electronegativity of nickel on the Pauling scale is 1.91, while the electronegativity of copper and silicon is 1.9. As a result, there is a shift of electron pairs from silicon and copper to nickel. This leads to the flow of a negative charge from copper and silicene to nickel, and the resulting layers of opposite charges create additional electrostatic adhesion between silicene and nickel and between nickel and copper.

Further remarks:

3) Line 159: “The geometric structure of silicene on copper and nickel substrate with fixed lower substrate layer are presented in Fig.2”. Is it means that the choice of the lattice in-plane (i.e., in the plane parallel to the surface) the lattice parameter of the substrate is keep fixed? what is its value? how was it chosen?

Fixing was carried out only in the z direction of the lower copper/nickel layer. In the xy plane, free movement of atoms occurred. The distance between the layers is consistent with the cubic face-centered cell of these elements in the 111 direction. Corrections have been made to the text:

The geometric structures of silicene on copper and nickel substrate with a fixed in z direction lower substrate layer are presented in Figure 2.  The distance between the layers (2.25 Å) is consistent with the cubic face-centered cell of these elements in the 111 direction.

4) In the introduction the authors stated: “copper has a high diffusion capacity and good solubility in Si” (line 29).” To prevent direct contact of Cu with the underlying Si, a thin layer of Ni is applied on the contact pads between Si and Cu.” (lines 32,33). This motivated the study in the paper of silicene system on cupper substrate with an intermediate layer of nickel. However, the authors found “significant structural changes observed in silicene in ab initio MD modeling on a nickel substrate” (line 263) for the silicene sheet and a nickel layer on copper substrate. In view of these finding, can the authors explain how they can conclude that “…the work indicates the possibility of effective use of silicene … for a silicene/copper solar cells with an intermediate nickel layer used as a diffusion barrier” (line 317). If silicene degrade on a Ni monolayer on Cu(111) it seems not possible to use silicene in such a system.

The system of silicene on a copper substrate with an intermediate nickel layer is stable in the considered temperature range. Corrections have been made to the text:

Appendix B shows simulations of silicene on a one-, two-, and three-layer copper substrate with an intermediate layer of nickel in between. Such systems are stable at a temperature of 293 K.

Round 2

Reviewer 3 Report

The authors have satisfactory accomplished all referee’s remarks. The paper can be published in Materials in the present form.